

# Overexpression of KIAA1199 is an independent prognostic marker in laryngeal squamous cell carcinoma

Meixiang Huang[1], Feifei Liao[2], Yexun Song[1], Gang Zuo[3], Guolin Tan[1], Ling Chu[2] and Tiansheng Wang[1]

[1] Department of Otolaryngology Head and Neck Surgery, The Third Xiangya Hospital, Central South University, Changsha, Hunan, China
[2] Department of Pathology, The Third Xiangya Hospital, Central South University, Changsha, Hunan, Chian
[3] Ministry of Education (Central South University), Key Laboratory of Metallogenic Prediction of Nonferrous Metals and Geological Environment Monitoring, Changsha, Hunan, China

## ABSTRACT

**Background:** KIAA1199 is a recently identified novel gene that is upregulated in various human cancers with poor survival, but its role and the underlying mechanisms in laryngeal squamous cell carcinoma (LSCC) remain unknown. Here, we collected tissues from 105 cases of LSCC to investigate the relationships between KIAA1199 protein expression and clinical factors.

**Methods:** Western blotting and real-time quantitative PCR (RT-PCR) were used for detect the protein and mRNA expression of KIAA1199 in LSCC tissue. Immunohistochemistry (IHC) staining was used to detect the expression of KIAA1199. Patient clinical information, for instance sex, age, pathological differentiation, clinical region, T stage, N stage, clinical stage, operation type, neck lymph dissection, smoking status, and drinking status were recorded. Kaplan–Meier survival analysis and Cox analysis were applied to identify the relationship between KIAA1199 and LSCC.

**Results:** Western blotting results showed KIAA1199 protein was significantly higher in tumor tissues vs. adjacent non-cancerous tissues ($0.9385 \pm 0.1363$ vs. $1.838 \pm 0.3209$, $P = 0.04$). The KIAA1199 mRNA expression was considerably higher in tumor tissues ($P < 0.001$) than in adjacent non-cancerous tissues by RT-PCR. IHC results showed up-regulated KIAA1199 expression was related with some severe clinicopathological parameters: pathologic differentiation ($P = 0.002$), T stage ($P < 0.001$), N stage ($P < 0.001$), clinical stage ($P < 0.001$), survival time ($P = 0.008$) and survival status ($P < 0.001$). Kaplan–Meier survival analysis showed that patients with high KIAA1199 protein expression had poor overall survival (OS) ($P < 0.05$). Cox analysis suggested that the KIAA1199 protein expression constituted an independent prognostic marker for LSCC patients ($P < 0.001$).

**Conclusion:** Our findings revealed that KIAA1199 protein expression may be used to predict LSCC patient outcome.

Corresponding authors
Ling Chu, 1799417560@qq.com
Tiansheng Wang, tswheaven@hotmail.com

## INTRODUCTION

Laryngeal squamous cell carcinoma is the most common laryngeal cancer, and it is the second highest incidence among head and neck cancer (*Iizuka et al., 2015*). In 2016, Laryngeal malignancies accounted for approximately 13,400 cancer cases in USA, of which an estimated 3,600 patients succumbed to the disease (*Siegel, Miller & Jemal, 2019*). The number of laryngeal cancer has radically changed in the last 20 years in USA (*Rothenberg & Ellisen, 2012*). In the available treatments, there have been improvements, but the patients still need identified novel gene that is, up-regulated in human cancer suffer from poor prognosis. So, in the pathogenesis of LSCC, identification of the key molecules is urgently needed to improve the treatment of LSCC.

The KIAA1199 gene was first discovered in association with non-syndromic hearing loss (*Abe, Usami & Nakamura, 2003*). Nowadays, we have known that the KIAA1199 gene is expressed in a wide range of normal human tissues (*Zhang, Jia & Weng, 2014*). Over-expression of KIAA1199 contributes to resistance to cell immortalization and cancerization in normal human cells and is associated with cell death (*Michishita et al., 2006*). Several researches have illuminated that KIAA1199 is over-expressed in different cancers, including oral squamous cell carcinoma (*Chanthammachat et al., 2013*), breast cancer (*Evensen et al., 2013*), gastric cancer (*Matsuzaki et al., 2009*), colorectal tumors (*Birkenkamp-Demtroder et al., 2011*; *LaPointe et al., 2012*), prostate cancer (*Michishita et al., 2006*), ovarian cancer (*Shen et al., 2019*) and hepatocellular carcinoma (*Gu et al., 2018*; *Jiang et al., 2018*). These studies showed that KIAA1199 regulates the proliferation, migration, and invasion of colorectal tumors, prostate cancer, ovarian cancer and so on. At this time there has been no report about KIAA1199 expression in LSCC, and the clinical value and biological role of KIAA1199.

We implemented immunohistochemical detection of KIAA1199 protein expression to investigate the clinical significance of KIAA1199 and to detect whether it plays a key role in the progression of LSCC in 105 paired formalin-fixed and paraffin-embedded cancer and adjacent noncancer tissues obtained from patients with LSCC. In the end, we illuminated the clinicopathologic characteristics of LSCC patients was related to KIAA1199, which were statistically evaluated.

## MATERIALS AND METHODS

### Patient enrollment and arrange follow up

The research was implemented in the Department of ENT, the Third Xiangya Hospital, Central South University. We collected 10 pairs of fresh specimens and their matched adjacent non-cancerous specimens, which were from patients diagnosed with human laryngeal squamous cell carcinoma by pathological examination in February 2018. A total of 105 patients who had been performed curative resection for LSCC were registered from 2009 to 2014. Patients with recurrence of laryngeal cancer or multiple cancers were excluded. No anticancer therapy was given before surgery. Postoperative pathological examination of patients diagnosed with laryngeal squamous cell cancer. Patient clinical data such as sex, age, pathological differentiation, clinical region, T stage, N stage, clinical

stage, operation type, neck lymph dissection, smoking status, and drinking status were collected. To investigate the prognostic value of KIAA1199 in postoperative patients, we examined the overall survival rate (OS) of the LSCC patients. The average arrange follow up cycle was 54 months (5 months extent to 10 years). Prior to the start of the study, we obtained the written informed consent of all patients and the approval of The Institutional Review Board of Third Xiangya Hospital, Central South University in accordance with the Helsinki Declaration Guidelines (No. 2018-S084). All tissue samples were treated and anonymous in accordance with ethical and legal standards. The tumor stage was determined according to the tumor, lymph node, metastasis (TNM) grading of the *International Union Against Cancer (UICC, 2002)*.

## RNA extraction and real-time RT-PCR

According to the manufacturer's protocol, the total RNA was isolated from LSCC and matched adjacent tissues by using TRIzol Reagent (Invitrogen, Waltham, MA, USA). Nanodroplet spectrophotometer (Thermo Scientific, Waltham, MA, USA) was used to measure the concentration and purity of total RNA. According to the manufacturer's instructions, the total RNA was converted to cDNA using a quantitative PCR (qPCR) reverse transcription kit (TOYOBO Life Science, Shanghai, PR China), fresh tissues were used to synthesize cDNA. Real-time RT-PCR was applied three times using a KOD SYBR qPCR Mix Fluorescent Quantitative PCR kit (TOYOBO Life Science, Shanghai, PR China). PCR and data collection were conducted by using an EP Real-time PCR System (Eppendorf Inc., Hauppauge, NY, USA). For standardization, we used GAPDH as an endogenous control. The primers used in our study were purchased from Sangon Biotech (Shanghai, PR China), and the following primer sequences were used: KIAA1199, F primer 5′-CCAGTAACCTGCGAATGAAGA-3′ and R primer 5′-TGGTCCCAGTGGATGGTGTAG-3′. GAPDH, F primer 5′-TTGGTATCGTGGAAGG ACTCA-3′ and R primer 5′-TGTCATCATATTTGGCAGGTT-3′. The reaction conditions were 95 °C for 5 min, followed by 40 cycles at 95 °C for 15 s and 58 °C for 30 s. The relative expression level was determined by the $2^{-\Delta\Delta Ct}$ method.

## Western blotting analysis

Proteins were extracted from LSCC fresh tissue samples and adjacent non-cancerous fresh tissue samples. The Western blotting analysis was carried out according to our previous article (*Li et al., 2012*). Primary antibodies were used as follows: polyclonal rabbit anti-KIAA1199 antibody (diluted 1:1,000), anti-GAPDH antibody (diluted 1:5,000), and horseradish peroxidase-conjugated secondary antibody (1:10,000).

## Immunohistochemistry

One hundred and five formalin-fixed, paraffin-embedded LSCC tissues were used for the immunohistochemistry (IHC) studies. Briefly, the tissue was sliced continuously into approximately four μm section, paraffin was removed from the sections using a graded alcohol series of 100% and 95% in xylene, rehydrated in 75%, and finally washed with PBS. Subsequently, the antigen was prepared with sodium citrate buffer PBS and incubated in

3% H2O2 deionized water for 15 min to inactivate endogenous peroxidase. The sections were washed three times with PBS, incubated with calf serum to block non-specific antigen for 10 min, incubated with polyclonal rabbit anti-KIAA1199 antibody (1:70) at room temperature for 1 hour, washed with PBS three times, and then incubated with secondary antibody at room temperature for 30 min. Sections were washed with PBS three times, stained with diaminobenzidine (DAB) for 4 min, washed three more times with PBS, restained with hematoxylin for 30 s, washed with flowing water, dried and sealed. Dried sections were observed with an optical microscope. The positive control was gastric cancer tissue confirmed by pathological examination, and the adjacent normal tissues from patients with LSCC were used as the negative control.

The positive expression of KIAA1199 was patchy with aggregates of brown granules in the cytoplasm. Semiquantitative analysis was used to determine the percentage of positive cells under the microscope and score the staining intensity. Two senior pathologists of the Department of Pathology were assigned to read the slides in a double-blinded manner, and 3–5 different fields were randomly selected from each IHC staining section for observation. The staining results were semiquantitatively analyzed in terms of staining intensity and percentage of cells with positive expression. Evaluation of dyeing intensity: range, 0–3; colorless (negative) = 0, weak (pale yellow) = 1, medium (brown–yellow) = 2, strong (tan) = 3. Percentage of stained cells: range, 0–3; percent positive cells <5% = 0, 0.5–10% = 1, 10–50% = 2, ≥50% = 3. The score of the two was multiplied to show the positive grade: 0 is negative (−), ≤3 is low expression, and >3 is higher expression.

### Statistical analysis

Our results were interpreted with GraphPad Prism version 7.0 (GraphPad Software, Inc., La Jolla, CA, USA) and SPSS 23.0 software package (SPSS, 112 Y.-H. HAO ET AL. Chicago, IL, USA). Chi-square test was used to analyze the associate with KIAA1199 protein expression and clinicopathological characteristics in LSCC patients. Cox regression analysis estimated the risk of death associated with KIAA1199 protein expression. Kaplan–meier method was used to analyze the total survival curve. Other data were analyzed using Student's $t$-test, and ANOVA was conducted to determine the differences in two or more groups. All data are presented as the mean ± SD with $P < 0.05$ ($^*P < 0.05$, $^{**}P < 0.01$, $^{***}P < 0.005$, $^{****}P < 0.001$).

## RESULTS

### Clinical data

To understand the clinical features of patients, the detailed data of the patients, such as sex, age, pathological differentiation, clinical region, T stage, N stage, clinical stages, operation type, neck lymph dissection, smoking status and drinking status, were collected from their medical records, and these data are summarized in Table 1 for the 105 patients in this study; 103 (98.1%) were men, and 2 (1.9%) were women, ranging in age from 37 to 82 years. T1–T2 stage was detected in 70 patients (66.6%), N1–N3 stage was detected in 20 patients (19%), and the OS time ranged from 6 to 108 months.

**Table 1 Clinicopathological characteristics of patient samples and expression of KIAA1199 in LSCC.**

| Parameters | | Case number/$n$ (%) |
|---|---|---|
| Gender | Male | 103 (98.1) |
| | Female | 2 (1.9) |
| Age (year) | ≤60 | 44 (41.9) |
| | >60 | 61 (58.1) |
| Pathologic differentiation | Poorly | 20 (19.05) |
| | Moderately | 29 (27.62) |
| | Highly | 56 (53.33) |
| Clinic Region | Supraglottic type | 10 (9.52) |
| | Trans glottic type | 5 (4.76) |
| | Glottic type | 87 (82.86) |
| | Subglottic type | 3 (2.86) |
| T stage | T1–T2 | 70 (66.6) |
| | T3–T4 | 35 (33.4) |
| N stage | N0 | 85 (81) |
| | N1–N3 | 20 (19) |
| Clinical Stages | I | 52 (49.5) |
| | II | 15 (14.3) |
| | III | 11 (10.5) |
| | IV | 27 (25.7) |
| Operation | Total laryngectomy | 28 (26.7) |
| | The partial laryngetomy | 77 (73.3) |
| Neck lymph dissection | No | 40 (38.1) |
| | Radical cervical clearing | 26 (24.8) |
| | Selective/functional neck cleanser | 39 (37.1) |
| Smoke | No | 30 (28.6) |
| | Yes | 75 (71.4) |
| Drink | No | 53 (50.5) |
| | Yes | 52 (49.5) |
| Expression of KIAA1199 | Low expression | 50 (47.6) |
| | High expression | 55 (52.4) |

## Increased expression of KIAA1199 in human LSCC tissues

To uncover the role of KIAA1199 expression in LSCC, we first detected KIAA1199 protein and mRNA expression in 10 pairs of fresh human LSCC specimens and their matched adjacent noncancerous specimens using Western blotting (Figs. 1A and 1B), IHC (Fig. 2) and RT-PCR (Fig. 1C). As shown in, KIAA1199 protein levels were significantly higher in LSCC tissues (1.838 ± 0.3209 vs. 0.9385 ± 0.1363, $P = 0.04$) (Fig. 1B) than in adjacent noncancerous tissues. RT-PCR revealed that KIAA1199 mRNA expression was considerably lower in adjacent noncancerous tissues ($P < 0.001$) than in cancer tissues (Fig. 1C). Then, we compared the expression of KIAA1199 in 105 LSCC tissues and their adjacent noncancerous tissues through IHC. There was weak or negative expression of

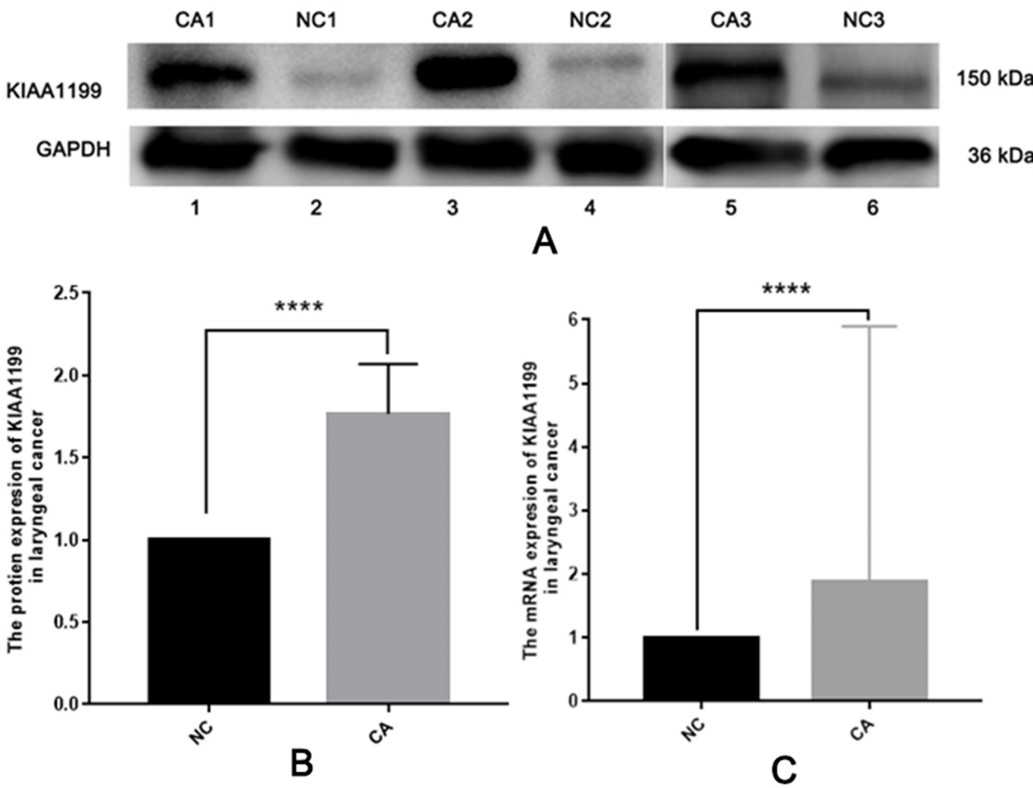

**Figure 1 The protein of KIAA1199 was overexpression in LSCC tissue specimens.** (A) and (B) The protien expression of KIAA1199 in adjacent non cancerous tissue and LSCC tissue by Western bloting. ****$P < 0.001$. (C) The mRNA expression of KIAA1199 in adjacent non cancerous tissue and LSCC tissue by RT-PCR. ****$P < 0.001$.

KIAA1199 in adjacent noncancerous tissue but high expression in the cytoplasm of LSCC tissue cells. The positive staining and negative staining rates in LSCC tissues were 52.4% (55/105, Table 1) and 47.6% (50/105), respectively. Semiquantitative analysis showed that KIAA1199 was significantly increased in LSCC tissues. Representative photographs of the immunostaining are shown in Figs. 3A–3F. RT-PCR results for KIAA1199 mRNA levels were agree with the Western blotting and IHC results, showing that KIAA1199 is increased in LSCC tissues.

## KIAA1199 expression is associated with pathologic differentiation, T, N, clinical stage, survival status and survival times of LSCC

In order to further reveal the character of KIAA1199 in LSCC, we evaluated the relationship between its expression and the clinicopathological characteristics of LSCC. As shown in Table 2, upregulation of KIAA1199 expression was associated with some clinicopathological parameters: pathologic differentiation ($P = 0.002$), T stage ($P < 0.001$), N stage ($P < 0.001$), clinical stage ($P < 0.001$), survival time ($P = 0.008$) and survival status ($P < 0.001$). However, KIAA1199 expression was not correlated with age, sex, clinical region, smoking status, or drinking status.

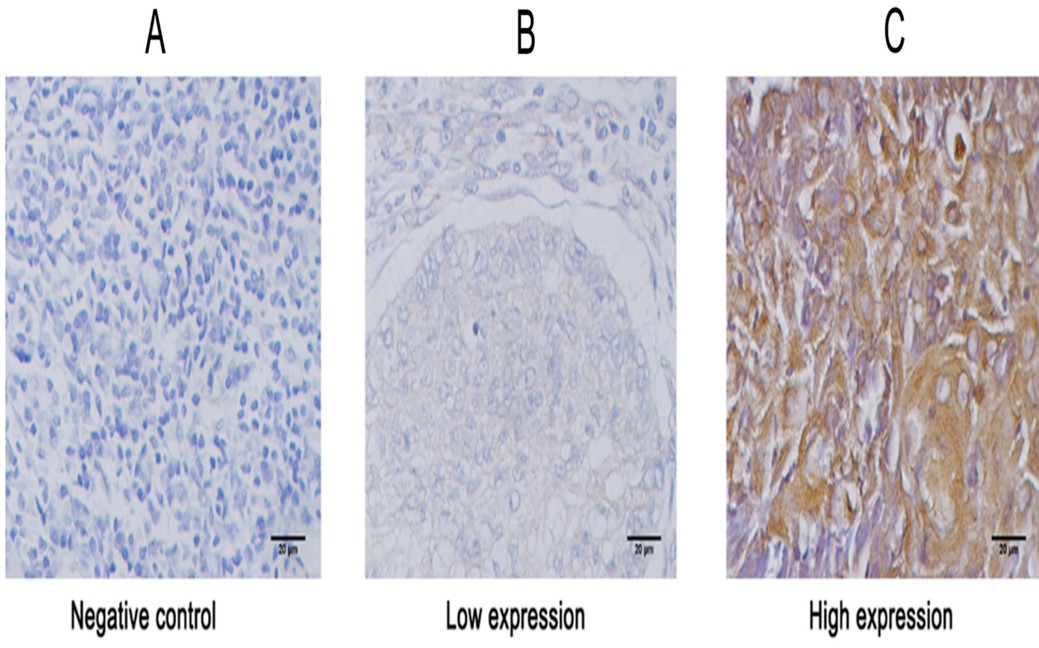

**Figure 2 Representative images of immunohistochemical staining for KIAA1199 expression in larynx specimens.** (A) Negative expression of KIAA1199 in adjacent non cancerous specimens. (B) Low expression of KIAA1199 in LSCC specimens. (C) High expression of KIAA1199 in LSCC specimens. Original magnification: 40×; scale bars: 20 um. 

## Survival assessment: a high level of KIAA1199 is predictive of poor prognosis in LSCC patients

The survival curve was plotted by kaplan–meier method, and the survival time was tested by log-rank test. The results showed that LSCC patients with high KIAA1199 expression had a lower prognosis, and low KIAA1199 expression in LSCC patients ($P < 0.001$, for OS) was related with considerably longer OS compared with high KIAA1199 expression. The median OS for high KIAA1199 expression was 60 ± 4.113 months and that for low KIAA1199 expression was 96 ± 7.928 months (Fig. 3G). Then analyzed independent prognostic factors for survival in patients with LSCC by using univariate and multivariate Cox proportional hazards analysis. The univariate analysis results (Table 3) showed that age (HR =1.032, 95% CI [1.001–1.063]; $P = 0.04$), pathologic differentiation (HR = 0.643, 95% CI [0.524–0.789]; $P < 0.001$), T stage (HR = 1.402, 95% CI [1.139–1.724]; $P < 0.001$), N stage (HR = 1.679, 95% CI [1.148–2.4577]; $P = 0.008$), clinical stage (HR = 1.445, 95% CI [1.180–1.769]; $P < 0.001$), operation type (HR = 0.380, 95% CI [0.222–0.650]; $P < 0.001$) and KIAA1199 expression (HR = 12.165, 95% CI [5.434–27.233]; $P < 0.001$) were significantly associated with the OS of LSCC patients. Multivariate survival analysis (Table 4) showed that KIAA1199 expression was statistically significant predictor of OS (HR = 27.937, 95% CI [10.600–73.632]; $P < 0.0001$) and that age (HR = 1.039, 95% CI [1.003–1.077]; $P = 0.0354$), clinical stage (HR = 0.704, 95% CI [0.581–0.960]; $P = 0.023$), operation type (HR = 0.285, 95% CI [0.093–0.870]; $P = 0.027$),

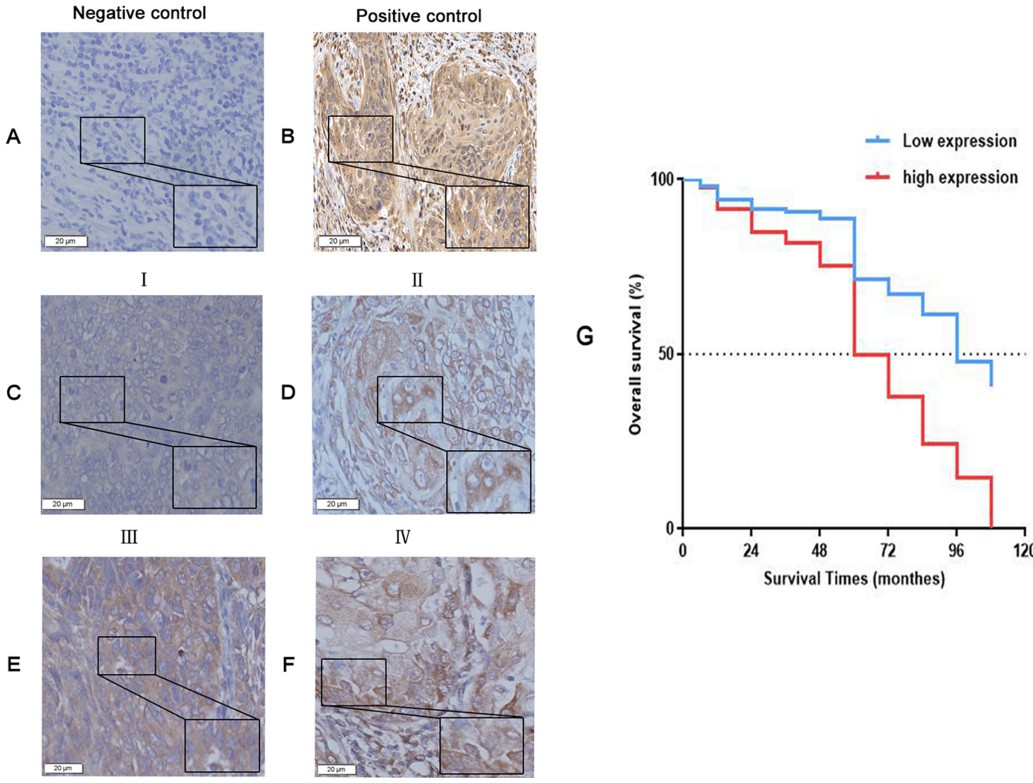

**Figure 3 The expression of KIAA1199 in LSCC tissues and survival surve.** (A–F) KIAA1199 expression by immunohistochemical staining. (A) Adjacent nonc-ancerous tissue as the negative control. (B) Gastric cancer tissue as the positive control. (C) I stage LSCC tissue. (D) II stage LSCC tissue. (E) III stage LSCC tissue. (F) IV stage LSCC tissue. (G) Kaplan–Meier survival curves analysis of ov-erall survival for all patients with KIAA1199 negative and positive LSCC tissue.

T stage (HR = 0.68, 95% CI [0.529–0.874]; $P$ = 0.003) and smoking status (HR = 0.19, 95% CI [0.057–0.630]; $P$ = 0.007) were independent predictive factors for OS.

## DISCUSSION

In order to identify novel gene that is, up-regulated in human cancer with poor prognosis, a deeply cognition to the molecular biology profiles of LCSS is a vital work. To outcomes remain elusive, the molecular pathways involved in LSCC incidence, progression and clinical yet. Espeially the endoplasmic reticulum, KIAA1199, which is a glycosylated protein which located in the cytoplasm and membrane (*Tiwari et al., 2013*; *Michishita et al., 2006*; *Evensen et al., 2013*). The relationship between cancer and KIAA1199 have been studied in many research directions. KIAA1199 is a recently identified novel gene that can regulate cell growth and invasion and could be a new therapeutic target in breast cancer (*Jami et al., 2014*). A similar analysis reported that KIAA1199 overexpression can predict poor survival in patients with colon cancer (*Xu et al., 2015*). By several mechanisms, KIAA1199 protein can accelerate cancer progression. Simultaneously, other research have shown that the KIAA1199 protein expression level is elevated upon p53 activation (*Matsuzaki et al., 2009*). KIAA1199 is also

**Table 2 Correlation between KIAA1199 expression and clinicopathologic characteristics of LSCC patients.**

| Parameters | Expression of KIAA1199 (No.) | | P |
|---|---|---|---|
| | Low | High | |
| Gender | | | |
| Male | 48 | 55 | 0.224 |
| Female | 2 | 0 | |
| Age (year) | | | |
| ≤60 | 21 | 23 | 1.000 |
| >60 | 29 | 32 | |
| Pathologic differentiation | | | |
| Poorly | 1 | 19 | <0.001 |
| Moderately | 18 | 11 | |
| Highly | 31 | 25 | |
| Clinic region | | | |
| Supraglottic type | 3 | 7 | 0.072 |
| Trans glottic type | 0 | 5 | |
| Glottic type | 46 | 41 | |
| Subglottic type | 1 | 2 | |
| T stage | | | |
| T1–T2 | 48 | 22 | <0.001 |
| T3–T4 | 2 | 33 | |
| N stage | | | |
| N0 | 50 | 35 | <0.001 |
| N1–N3 | 2 | 18 | |
| Clinical Stage | | | |
| I–II | 48 | 19 | <0.001 |
| III–IV | 2 | 36 | |
| Smoke | | | |
| No | 18 | 12 | 0.132 |
| Yes | 32 | 43 | |
| Drink | | | |
| No | 26 | 27 | 0.846 |
| Yes | 24 | 28 | |
| Survival status | | | |
| Survive | 43 | 7 | <0.001 |
| Death | 7 | 48 | |
| Survival times (month) | | | |
| ≤12 | 2 | 7 | 0.008 |
| >12, ≤36 | 0 | 8 | |
| >36, ≤60 | 23 | 22 | |
| >60 | 25 | 18 | |
**Table 3 Univariate analyses of various prognostic parameters in patients with LSCC.**

| Parameters | Univariate Cox | | |
|---|---|---|---|
| | Hazard ratio | 95% CI | *P*-value |
| Gender | 0.048 | [0–25.791] | 0.422 |
| Age (year) | 1.032 | [1.001–1.063] | 0.040 |
| Pathologic differentiation | 0.643 | [0.524–0.789] | <0.001 |
| Clinic Region | 0.068 | [0.49–1.026] | 0.068 |
| T stage | 1.402 | [1.139–1.724] | 0.001 |
| N stage | 1.679 | [1.148–2.457] | 0.008 |
| Clinical Stages | 1.445 | [1.180–1.769] | <0.001 |
| Operation | 0.380 | [0.222–0.650] | <0.001 |
| Neck lymph dissection | 0.957 | [0.7106–1.291] | 0. 774 |
| Smoke | 1.028 | [0.560–1.885] | 0.930 |
| Drink | 0.782 | [0.460–1.330] | 0.365 |
| Expression of KIAA1199 | 12.165 | [5.434–27.233] | <0.001 |

**Table 4 Multivariate analyses of various prognostic parameters in patients with LSCC.**

| Parameters | Multivariate Cox | | |
|---|---|---|---|
| | Hazard ratio | 95% CI | *P*-value |
| Age (year) | 1.039 | [1.003–1.077] | 0.035 |
| Clinic Stage | 0.704 | [0.581–0.960] | 0.023 |
| Operation | 0.285 | [0.093–0.870] | 0.027 |
| T stage | 0.68 | [0.529–0.874] | 0.003 |
| Smoke | 0.400 | [0.204–0.785] | 0.008 |
| Expression of KIAA1199 | 27.937 | [10.600–73.632] | 0.001 |

related to angiogenesis in rheumatoid arthritis (*Yang et al., 2015*). However, the mechanism of KIAA1199 tumor-promoting effects in LSCC is little known.

In our study, we first verified KIAA1199 protein and mRNA expression in 10 pairs of fresh surgically resected LSCC samples by Western blotting, IHC and real-time RT-PCR. Our results have drawn a conclusion that KIAA1199 was highly expressed in LSCC cancerous in contrast to adjacent non-cancerous tissue. In view of our data, we also can censor the hidden expression of KIAA1199 by IHC in 105 paraffin-embedded sections (2009–2014) to further explore the relationship between KIAA1199 and clinicopathological characteristics. Our data analysis showed that KIAA1199 expression was not kenspeckle related with clinical parameters which as age, sex, clinical region, smoking, or drinking. Interestingly, for some severe clinicopathological parameters: pathologic differentiation ($P = 0.002$), T stage ($P < 0.001$), N stage ($P < 0.001$), clinical stage ($P < 0.001$), survival time ($P = 0.008$) and survival status ($P < 0.001$), the significant correlations were observed. Through our experiments, we obtained many data, which

provides a new evidence that KIAA1199 is highly expressed in primary LSCC tissues and its immunoreactivity is higher in cancerous than adjacent noncancerous tissues, revealing that KIAA1199 might help distinguish benign from malignant larynx tumors. Moreover, our results and analysis illuminated that the expression of KIAA1199 was elevated in LSCC tissues with aggressive clinicopathological characteristics, suggesting its potential as a marker of cancer invasionality.

The abnormal expression of KIAA1199 has also been found in other cancer studies, such as oral squamous cell carcinoma (*Chanthammachat et al., 2013*), breast cancer (*Evensen et al., 2013*), gastric cancer (*Matsuzaki et al., 2009*), colorectal tumors (*Tiwari et al., 2013*; *Birkenkamp-Demtroder et al., 2011*; *LaPointe et al., 2012*), prostate cancer (*Michishita et al., 2006*), ovarian cancer (*Shen et al., 2019*) and hepatocellular carcinoma (*Jiang et al., 2018*). It was reported (*Jiao et al., 2019*) that KIAA1199 was abnormaly increased in the papillary thyroid tumor compared with normal specimens tissues and that upregulation of KIAA1199 was positively correlated with more advanced clinical variables. There was analysis showed that the cell invasion and migration were related with KIAA1199. KIAA1199 silencing inhibited the invasive ability of papillary thyroid cancer cells by affecting epithelial-mesenchymal transition (EMT) in vitro and in vivo. Additionally, the same as our study, In clone cancer study (*Xu et al., 2015*) proved the expression of KIAA1199 was also observably associated with tumor invasion, metastasis and TNM staging. Increased mortality risks associated with overexpression of KIAA1199 in primary hepatocellular cancer patient. Previous researches have demonstrated that up-regulation of KIAA1199 motivates carcinogenesis, motility and apoptosis. Metastasis, invasion, and cell movement of a variety of cell types are associated with KIAA1199 expression (*Zhang, Jia & Weng, 2014*). By the Wnt/β-catenin signaling pathway, EMT is one of the important processes mediated, which plays a key role in cancer invasion and metastasis (*Wu et al., 2012*). Interestingly, the KIAA1199 signaling pathway also induces the development and progression of tumor. Other researches showed that the cell proliferation and mobility of colorectal cancer cells were inhibited by knocking down the expression of CEMIP in vitro, and the EMT process of colorectal cancer cells is suppressed by shRNA-CEMIP via inactivation of the Wnt/β-catenin/Snail pathway (*Liang et al., 2018*). Collectively, our results demonstrated that the overexpression of KIAA1199 mRNA may affect tumor spread, lymph node metastasis, tumor differentiation and prognosis (*Matsuzaki et al., 2009*). In a report, it was defined KIAA1199 as an carcinogenic protein induced by HPV infection and compositive NF-kB activity that transmits pro-survival and aggressive signals via EGFR signalling (*Shostak et al., 2014*). Research has suggested that KIAA1199 may promote the development of ovarian cancer by regulating PI3K/AKT signalling (*Shen et al., 2019*). One study insisted, AMPK/GSK3β/β-catenin cascade triggered KIAA1199 over-expression may promote migration and invasion in anoikis-resistant prostate cancer cells by increasing PDK4-associated metabolic reprograming, which may provide a novel therapeutic target for the prostate cancer (*Zhang et al., 2018*). Therefore, in the light of the upper research about the KIAA1199-related signaling pathway, we can draw a conclusion that KIAA1199 can influence the occurrence and development of laryngeal cancer, which may also be related to the

Wnt/β-catenin, EGFR, PI3K/AKT and AMPK/GSK3 signaling pathways and other pathways. Thence, we will carry out a molecular mechanism research of KIAA1199 in LSCC cells and animal models in our future study.

Some limitations exist in our research. First, the sample size of this study was a little small. As a retrospective study design that the selection bias might not be ignored. Second, our study did not explore the effect of other treatments for LSCC on the prognosis of patients, such as radiotherapy and chemotherapy. So, in the future studies, we will carry out cell biology experiments to verify our findings, such as gene transfection and cell migration assays.

## CONCLUSIONS

In conclusion, our results revealed significant associations of KIAA1199 protein expression with various clinicopathologic characteristics and the prognosis of LSCC patients. Moreover, survival analysis illuminated KIAA1199 was an independent prognostic factor for OS in LSCC. All of these findings indicate that the KIAA1199 protein might be used as a pathological marker to identify individuals with poor outcomes and to provide a reference for clinical therapy in the future. Further studies are required to investigate its rationality as a marker and the potential pathways involved in KIAA1199-mediated cell invasion and metastasis.

## ACKNOWLEDGEMENTS

We thank Dr. Sheng Xiao for assistance with the IHC experiments.

### Funding
This work was supported by the Hunan Provincial Science Foundation of China (No. 2016JJ4104) and the National Natural Science Foundation of China (No. 81702706). The funders had no role in study design, data collection and analysis, decision to publish, or preparation of the manuscript.

### Grant Disclosures
The following grant information was disclosed by the authors:
Hunan Provincial Science Foundation of China: 2016JJ4104.
National Natural Science Foundation of China: 81702706.

### Competing Interests
The authors declare that they have no competing interests.

### Author Contributions
- Meixiang Huang performed the experiments, analyzed the data, prepared figures and/or tables, and approved the final draft.
- Feifei Liao performed the experiments, prepared figures and/or tables, and approved the final draft.

- Yexun Song analyzed the data, authored or reviewed drafts of the paper, and approved the final draft.
- Gang Zuo analyzed the data, authored or reviewed drafts of the paper, and approved the final draft.
- Guolin Tan analyzed the data, authored or reviewed drafts of the paper, and approved the final draft.
- Ling Chu conceived and designed the experiments, authored or reviewed drafts of the paper, and approved the final draft.
- Tiansheng Wang conceived and designed the experiments, authored or reviewed drafts of the paper, and approved the final draft.

## Human Ethics

The following information was supplied relating to ethical approvals (i.e., approving body and any reference numbers):

Institutional Review Board of Third Xiangya Hospital, Central South University granted ethical approval to carry out the study within its facilities (No. 2018-S084).

## Data Availability

The raw measurements are available as Supplemental Files.

## Supplemental Information

Supplemental information for this article can be found online at http://dx.doi.org/10.7717/peerj.9637#supplemental-information.

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
