# Peer review of "Overexpression of KIAA1199 is an independent prognostic marker in laryngeal squamous cell carcinoma"

_PeerJ, doi:10.7717/peerj.9637_

## Round 0.1 · original submission · Major Revisions

All the issues pointed by both reviewers should be addressed and the manuscript should be revised accordingly.

Reviewer 1 ·

Basic reporting

1. The writing of this manuscript is far from satisfactory. The authors need to find someone good at English to correct all the grammatical errors and misuse of terms including but not limited to sentences with missing verbs (e.g. Line 102), excessive use of clause (e.g. Line 246), and sentences making no sense at all (e.g. Line 245).
2. The authors need to define some of the explanatory variables used in their statistical analysis such as “Pathological Differentiation” and “Clinical Stage”. They need to briefly explain what these factors are, why they are included in their analysis, what are the levels of these factors, etc. They also need to describe by what standard do they classify someone as being drinking, i.e. how frequent does one need to drink to be classified as so.
3. Writings between Line 76 and Line 84 are contradictory. On one hand, the authors claim that “over-expression of KIAA1199 contributes to resistance to cell immortalization and cancerization” (Line 76-78). In the next sentence, they say “KIAA1199 is over-expressed in different cancers”. Does it means that over-expression of KIAA1199 is a part of our body’s defense mechanism to prevent cancer cells from becoming more malignant and invasive? If so, higher expression level of KIAA1199 should be beneficial instead of being correlated with low survival rate as suggested by the results of this paper.
4. There are some missing citations such as in Line 109, it needs a reference to the Helsinki Declaration. Line 141, a reference to the original immunohistochemistry protocol adopted by the authors. Line 173-174, reference to the original (or the version adopted by the statistical software used in this report) Cox regression and Kaplan-Meier survival analysis.
5. In Line 149, spell out the abbreviation DAB.
6. Wrong number in Line 202, “… in LSCC tissues were 52.4%(54/105, Table 1) …”. According to Table 1, this number should be 55/105.
7. In all tables, the levels of the factor “Clinical Stages” do not show up properly.
8. The authors use the term “independent predictor” several times (such as in Line 234). This should be changed to “statistically significant predictor”.
9. The figure legends need to be moved from figures to the end of the main text.

Experimental design

1. What is the sample size for the RT-PCR and western blot data shown in Figure 1? 10 or 105? If it were ten, how did the authors choose which samples to use?
2. Why did the authors exclude patients with recurrent and multiple cancers?
3. How did the authors select the 105 patients other than excluding cases with recurrent and multiple cancers?
4. The number of female patients is too small (only two) to make any valid statistical inference. Should just remove gender as a potential predictor in all statistical analysis reported here.
5. In Line 101 “No anticancer therapy was given before surgery”. Is it ethical to deny patients with anticancer therapy before surgery? Is it out of consideration of achieving the best treatment results or just for the sake of this study?
6. The authors conducted a series of Chi square association tests to evaluate the relationship between expression level of KIAA1199 and other factors. For the results to be valid, the authors need to check chi-square test assumptions regarding the underlying distribution of the data. Because the chi-square test is based on asymptotic approximation with normal distribution, it is generally required that for any entry of the n*m way table, the expected count should be at least five. If this assumption were not valid, the authors need to consider alternative non-parametric methods to test for association such as the Fisher’s exact test.
7. The authors need to check the assumptions of using Cox proportional hazards analysis: one such assumption is that the hazardous ratio is assumed to remain constant over time. This can be visualized by plotting Kaplan-Meier survival curves for each of the categorical variable. Paralleling survival curves are a good indicator of constant hazardous ratio.
8. Overall survival rate (OS) is usually complicated by competing events such as death not due to the disease of interest. I wonder how did the authors deal with such kind of complication?

Validity of the findings

1. The hazardous ratios reported in the paper lack information of the base level of the corresponding factor. Without it, it is hard to evaluate how those significant factors affect the survival time.
2. The error bars for NC are missing in Figure 1 B&C. Moreover, it is hard to believe comparison shown in Panel C is significant given the huge error bar associated with CA.
3. There are a few counter-intuitive points drawn from the results. One is that as shown in Table 3&4, smoke is a non-significant predictor for the survival time of these 105 patients. This is in contrast to previously published results such as this one: ORL J Otorhinolaryngol Relat Spec. 2012;74(5):250-4.
4. Another one is that one learns from Table 2 that the expression level of KIAA1199 appears to change from being largely low (48 patients out of 70) in T1/T2 stage to largely high in T3/T4 stage (33 out of 35). And then, it drops to largely low (50 out of 85) in N0 stage and rises back up to predominantly high (18 out of 20) in N1-N3. The fluctuation of the expression level appears to be very unusual. Could the authors comment on it?

Additional comments

NA

Reviewer 2 ·

Basic reporting

The experiments are solid, and the observations are very clear.

Experimental design

No comments

Validity of the findings

No comments

Additional comments

There are several grammatical and minor spelling errors in the manuscript, which need to be improved

Annotated reviews are not available for download in order to protect the identity of reviewers who chose to remain anonymous.

Reviewer 3 ·

Basic reporting

In this study, the author explored the relationship between KIAA1199 expression and laryngeal squamous cell carcinoma (LSCC) progression using Western blotting, RT-PCR and IHC. However, the roles of KIAA1199 in other cancer types have been extensively discussed, like oral squamous cell carcinoma (Pitak Chanthammachat, et al., 2013), breast cancer (Nikki A. Evensen, et al., 2013), gastric cancer(Shinji Matsuzaki, et al., 2009), colorectal tumors (K Birkenkamp-Demtroder, et al., 2011; Lawrence C. LaPointe, et al., 2012), prostate cancer (Eriko Michishita, et al., 2006), ovarian cancer(Fan Shena, et al., 2019) and hepatocellular carcinoma (Zhengchen Jiang, et al., 2018), which renders the shortage of novelty of the present study. The author should conduct some mechanistic studies to explore how KIAA1199 participates in LCSS, affecting cell proliferation, invasion or migration? Other than novelty, the language issues were spread in the whole manuscript, which should be corrected before submitted to scientific journals, the author should seek help from academic editors or scientific researches with profound writing experience. The detailed language issues were listed below.

Why does this study recruit only 2 females, but 103 males, it may cause gender bias of the study.
Real time RT-PCR is wrong written, should be real time quantitative PCR (RT-PCR).
“Western blotting” or “Western blot”, please be consistent throughout the manuscript.
In Abstract, the author described “we collected tissues from 105 cases of
laryngeal squamous cell carcinoma (LSCC) to investigate the relationships between
KIAA1199 protein expression and clinical factors.” Did the author compare LSCC tissues with noncancerous tissues (control), what’s the number of control tissues?
Citation in the context is not consistent, blank should be placed between context with citation. Such as Line 66, 68, 69. All of these should be revised throughout the manuscript.
The writing from Line 70 to line 71 should be rephrased.
Line 75, “Nowaday” should be changed to “Nowadays”.
Line 95 “arrange” and “follow-up” should be separated.
In line 99, A total of 105 patients were recruited, how about the control tissues and where are they from, please specifically identify in the section of Materials.
In line 97, “We collected 10 pairs of fresh specimens and their matched adjacent
98 noncancerous specimens”, what’s the specific specimens were collected, please list the details.
In line 140, for IHC, what’s the control tissue?
In line 155, the section of “Immunohistochemical staining results” is redundant, please combine with subtitle “Immunohistochemistry”.
In line 223, what’s the meaning of “OS”? When it appears at the first time, please give the full name and designate the abbreviation.
Comma is not corrected used in many cases, like line 253.
Line 259, “draw” should be replaced with “drawn”.
Line 316, “We” should be replaced with “we”.

Experimental design

Why does this study recruit only 2 females, but 103 males, it may cause gender bias of the study.

Validity of the findings

In this study, the author explored the relationship between KIAA1199 expression and laryngeal squamous cell carcinoma (LSCC) progression using Western blotting, RT-PCR and IHC. However, the roles of KIAA1199 in other cancer types have been extensively discussed, like oral squamous cell carcinoma (Pitak Chanthammachat, et al., 2013), breast cancer (Nikki A. Evensen, et al., 2013), gastric cancer(Shinji Matsuzaki, et al., 2009), colorectal tumors (K Birkenkamp-Demtroder, et al., 2011; Lawrence C. LaPointe, et al., 2012), prostate cancer (Eriko Michishita, et al., 2006), ovarian cancer(Fan Shena, et al., 2019) and hepatocellular carcinoma (Zhengchen Jiang, et al., 2018), which renders the shortage of novelty of the present study. The author should conduct some mechanistic studies to explore how KIAA1199 participates in LCSS, affecting cell proliferation, invasion or migration?

Additional comments

In this study, the author explored the relationship between KIAA1199 expression and laryngeal squamous cell carcinoma (LSCC) progression using Western blotting, RT-PCR and IHC. However, the roles of KIAA1199 in other cancer types have been extensively discussed, like oral squamous cell carcinoma (Pitak Chanthammachat, et al., 2013), breast cancer (Nikki A. Evensen, et al., 2013), gastric cancer(Shinji Matsuzaki, et al., 2009), colorectal tumors (K Birkenkamp-Demtroder, et al., 2011; Lawrence C. LaPointe, et al., 2012), prostate cancer (Eriko Michishita, et al., 2006), ovarian cancer(Fan Shena, et al., 2019) and hepatocellular carcinoma (Zhengchen Jiang, et al., 2018), which renders the shortage of novelty of the present study. The author should conduct some mechanistic studies to explore how KIAA1199 participates in LCSS, affecting cell proliferation, invasion or migration? Other than novelty, the language issues were spread in the whole manuscript, which should be corrected before submitted to scientific journals, the author should seek help from academic editors or scientific researches with profound writing experience. The detailed language issues were listed below.

Why does this study recruit only 2 females, but 103 males, it may cause gender bias of the study.
Real time RT-PCR is wrong written, should be real time quantitative PCR (RT-PCR).
“Western blotting” or “Western blot”, please be consistent throughout the manuscript.
In Abstract, the author described “we collected tissues from 105 cases of
laryngeal squamous cell carcinoma (LSCC) to investigate the relationships between
KIAA1199 protein expression and clinical factors.” Did the author compare LSCC tissues with noncancerous tissues (control), what’s the number of control tissues?
Citation in the context is not consistent, blank should be placed between context with citation. Such as Line 66, 68, 69. All of these should be revised throughout the manuscript.
The writing from Line 70 to line 71 should be rephrased.
Line 75, “Nowaday” should be changed to “Nowadays”.
Line 95 “arrange” and “follow-up” should be separated.
In line 99, A total of 105 patients were recruited, how about the control tissues and where are they from, please specifically identify in the section of Materials.
In line 97, “We collected 10 pairs of fresh specimens and their matched adjacent
98 noncancerous specimens”, what’s the specific specimens were collected, please list the details.
In line 140, for IHC, what’s the control tissue?
In line 155, the section of “Immunohistochemical staining results” is redundant, please combine with subtitle “Immunohistochemistry”.
In line 223, what’s the meaning of “OS”? When it appears at the first time, please give the full name and designate the abbreviation.
Comma is not corrected used in many cases, like line 253.
Line 259, “draw” should be replaced with “drawn”.
Line 316, “We” should be replaced with “we”.

---

## Round 0.2 · Major Revisions

As you can see, reviewer #3 is very disappointed and claims that you ignored the questions raised by reviewers and failed to answer the critiques in the first round of review. Based on these issues, this reviewer considers that your manuscript is not publishable and recommended rejection. However, I decided to give you another chance to revise your manuscript.

Reviewer 1 ·

Basic reporting

NA

Experimental design

NA

Validity of the findings

NA

Additional comments

The authors have made some effort in addressing some of the concerns and questions raised by the three reviewers, and yet some other questions are left either uncommented or insufficiently addressed.

I would like to see the authors to address all the questions and concerns from Reviewer 1 regarding the validities of the findings which are left uncommented. In particular, I would like to have your comments on the first two questions which are directly related to the credibility of the reported data and conclusions drawn from them in this paper.

Response to the third reviewer’s questions seems to be quite haphazard making me wonder if the authors proofread the rebuttal letter before submitting it.

In addition to these obvious shortfalls, I also notice that for some questions, although changes and corrections were promised by the authors, they have not been incorporated into the current version of the manuscript. These include Q3 (Basic reporting, Reviewer #1) and Q6 (Experimental design, Reviewer #1).

For the sake of clarity and reproducibility, I urge the authors to incorporate the information they provided in addressing the reviewers’ questions, in particular, in Q2 (Basic reporting, Reviewer #1), Q2 (Experimental design, Reviewer #1), and Q3 (Experimental design, Reviewer #1).

Reviewer 3 ·

Basic reporting

The current manuscript fails to answer all of the questions raised by reviewers, and in the context of the response to reviewers, the author includes some characters not in English, which makes it totally unreadable. Thus, highly recommend to reject the paper for publishing on scientific journal.

Experimental design

The logic and questions raised in the first round of review regarding "Experimental design " haven't been corrected.

Validity of the findings

The validity of the findings is problematic, which has been explained in the first round of review. However, the author ignores the comments and fails to correct it.

Additional comments

The revised version of manuscript fails to answer all of the questions and concerns raised in the first round of peer-review. Thus, the current manuscript should be rejected for publication.

---

## Round 0.3 · accepted · Accept

Thank you for addressing the remaining issues and finalizing your manuscript. I am pleased to let you know that your manuscript is accepted now.